# Quality control of whole-slide images through multi-class semantic segmentation of artifacts

**Gijs Smit**[1]                                    GIJS.SMIT@RADBOUDUMC.NL

**Francesco Ciompi**[1]                             FRANCESCO.CIOMPI@RADBOUDUMC.NL

**Maria Cigéhn**[2]                                 MARIA.CIGEHN@REGIONOSTERGOTLAND.SE

**Anna Bodén**[2]                                   ANNA.C.BODEN@REGIONOSTERGOTLAND.SE

**Jeroen van der Laak**[1]                          JEROEN.VANDERLAAK@RADBOUDUMC.NL

**Caner Mercan**[1]                                 CANER.MERCAN@RADBOUDUMC.NL

[1] *Computational Pathology Group, Radboud University Medical Center, The Netherlands.*

[2] *Department of Clinical Pathology, and Department of Biomedical and Clinical Sciences, Linköping University, Sweden.*

**Editors:** Under Review for MIDL 2021

## Abstract

Quality control is an integral part of the digitization process of whole-slide histopathology images due to artifacts that arise during various stages of slide preparation. Manual control and supervision of these gigapixel images are labor-intensive. Therefore, we report the first multi-class deep learning model trained on whole-slide images covering multiple tissue and stain types for semantic segmentation of artifacts. Our approach reaches a Dice score of 0.91, on average, across six artifact types, and outperforms the competition on an external test set. Finally, we extend the artifact segmentation network to a multi-decision quality control system that can be deployed in routine clinical practice.

**Keywords:** Multi-class, artifact segmentation, quality control, digital pathology.

## 1. Introduction

The introduction of whole-slide imaging technology enabled the digitization of glass slides and computational analysis of histologic tissues through deep learning (DL) to support pathologists in routine clinical practice. However, artifacts that emerge during slide preparation and digitization processes can obstruct tissue parts to an extent that the accurate analysis of the slide becomes difficult for pathologists as well as for DL-based decision systems. This problem calls for automated quality control systems that can identify the root cause and the severity of the artifacts to propose solutions for a more accurate analysis of whole-slide images.

In this work, we present the first multi-class deep learning approach for the semantic segmentation of the artifacts caused by out-of-focus, tissue folds, ink, dust, marker, and air bubbles.[1] Our dataset includes multiple tissue and staining types, making our approach suitable for different organ pathology as well as stain variations. We show that our method achieves superior performance compared to its competition, HistoQC (Janowczyk et al., 2019). Additionally, it can be seamlessly extended into a quality control framework that can automatically inform about the required actions to fix issues caused by artifacts. Overview of the proposed framework is shown in Figure 1.

---

1. grand-challenge.org/artifact-segmentation, github.com/DIAGNijmegen/pathology-artifact-detection

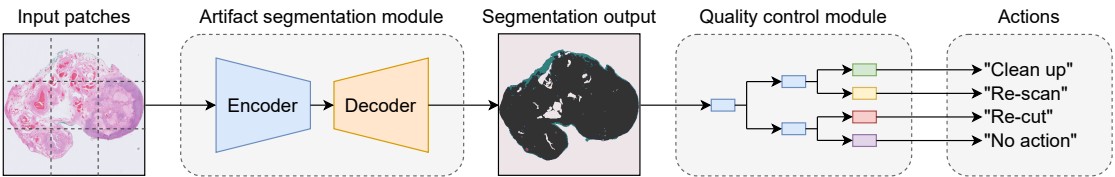

**Figure 1:** Overview of the proposed quality control framework.

## 2. Materials & Methods

**Datasets.** We collected a diverse set of 142 whole-slide images, consisting of 9 tissue and 8 stain types, digitized using 7 scanners. In total, we annotated 3,278 artifacts exhaustively in one or more regions selected from each slide. In our experiments, we split the dataset into a training (70%), validation (15%), and test (15%) set. Due to the gigapixel resolutions of the images, we sampled patches of $320 \times 320$ pixels at the spatial resolution of $4.00\,\mu m/pixel$, which were fed through our segmentation module during training. Additionally, we collected a set of 30 unannotated images from an external data center to evaluate the performance of our approach vs. the competition, which we refer to as external test set.

**Artifact segmentation module.** The proposed framework contained an artifact segmentation module, consisting of a DL network based on DeepLabV3+ (Chen et al., 2018). This architecture incorporated a spatial pyramid pooling module into an encoder-decoder network structure to improve the encoding of multi-scale contextual information. We used EfficientNet-B2 (Tan and Le, 2019) as the encoder of the network. Inference was comprised of two stages. First, we used a tissue segmentation network[2] to pre-process an input whole-slide image to eliminate the white spaces around the tissue block. Second, we applied the trained artifact segmentation network on the pre-processed image by feeding patches of $1024 \times 1024$ pixels at the spatial resolution of $4.00\,\mu m/pixel$ with test-time spatial augmentations of $90°$, $180°$, $270°$ rotations, as well as horizontal and vertical flips.

**Quality control module.** The framework contained a quality control module in the form of a decision tree to transform the artifact segmentation output into one of the four actions; clean up, re-scan, re-cut, and do nothing, for accurate analysis of input whole-slide images.

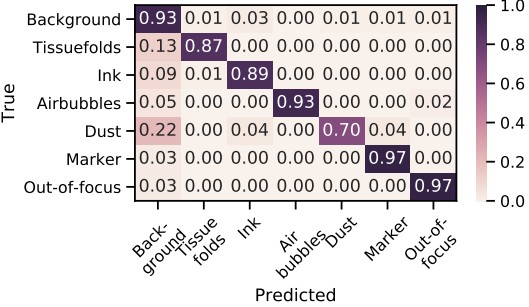

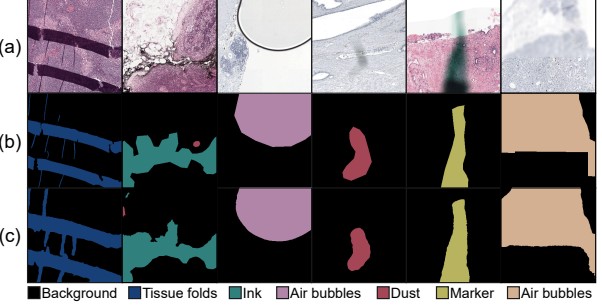

**Figure 2:** Pixel-level confusion matrix for classification of artifacts on the test set.

**Figure 3:** Qualitative results on the test set. (a) Image patches. (b) Ground truths. (c) Predictions.

2. doi:10.7717/peerj.8242

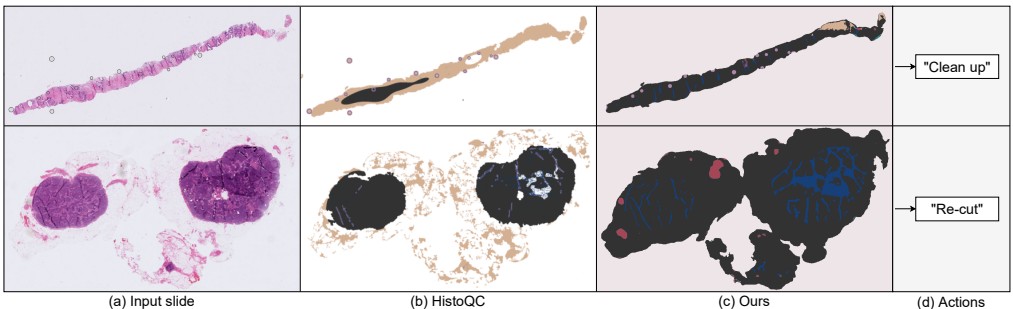

**Figure 4:** Comparison of (b) HistoQC vs. (c) our approach, including (d) quality control actions.

## 3. Results & Discussion

We computed pixel-level classification metrics to evaluate the quantitative performance of the artifact segmentation module, as presented in Figure 2. Additionally, our method achieved a patch-level Dice score of 0.91 on test images, averaged over the classes. The segmentation performance on air bubble, marker, and out-of-focus artifacts was the best in pixel- and patch-level evaluations. The relatively lower accuracy on dust artifacts was mostly due to the difficulty of annotating the boundaries of the artifact, which was evident from the false predictions as background (tissue region without artifact). Additionally, we demonstrate the qualitative results of our method on the test patches in Figure 3. Finally, we compared the slide-level qualitative performance of our artifact segmentation module vs. HistoQC on the external test set. We present this comparison, as well as the corresponding actions of our quality control module, for two example images in Figure 4.

## 4. Conclusion & Future Work

In this work, we proposed the first DL-based multi-class semantic segmentation of artifacts to perform quality control of whole-slide images in digital pathology. Our approach achieved high-precision artifact segmentation, and outperformed its competition on the external test set. In our future work, we will investigate DL-based quality control module alternatives.

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
