# OpenReview forum: "Quality control of whole-slide images through multi-class semantic segmentation of artifacts"
_MIDL.io/2021/Conference/Short — MIDL 2021 Poster_

### Official Review · Reviewer_7JwE · 2021-05-01

**Confidence:** 3
**Final Rating:** 4

**Summary:**

The authors present the first deep learning approach for semantic segmentation of the artifacts in whole slide histopathology images covering multiple tissues and stain types. Their contributions include (i) training a DL network based on DeepLabV3+ for performing this pixel-wise classification (ii) proposing a quality control module which produces a control action on the segmentation output through a decision tree procedure (iii) achieving superior performance compared to a baseline method HistoQC.

**Strengths:**

- First DL approach for semantic segmentation of artifacts
- The authors made the code public
- A quality control module is proposed thus making the proposed framework self-contained for deploying in routine clinical practice
- The authors report a high DICE score of 0.91 on artifact segmentation in the evaluation dataset


**Weaknesses:**

- The Materials and Methods section lacked details such as the number of training epochs, optimizer etc (were the default settings used from DeepLabV3?)
- The reported evaluation results were obtained from one run (no standard error in the DICE score is reported)

**Deanonymize Review:**

yes

**Detailed Comments:**

- Quantification of performance of baseline method HistoQC could be included (in addition to the qualitative comparison which was addressed by the authors)
- How does this DL network employed compare with standard Encoder-decoder architectures such as a multi-class Unet? (Quantification on additional baselines could be included)
- Training procedure of the quality control module could be included.


**Justification Of The Rating:**

The authors present a solution for artifact segmentation and quality control on whole slide histopathology images covering multiple tissues and stain types. Their solution is novel and achieves a high DICE score for segmenting artifacts, suggesting that their method works well. Furthermore, the authors provide a quality control module based on a random forest, which suggest actions needed to be taken by the user. In this way, their proposed framework can be deployed to automate quality control in clinical practice. I found the text well written and the discussion around the qualitative predictions engaging. Furthermore the authors make their code publicly available. Hence, I rate this work as a 'strong accept'.


**Paper Type:**

validation/application paper

**Special Issue:**

no

---

### Meta-Review · Area_Chair_Ab88 · 2021-05-11

**Recommendation:** Accept (Poster)
**Confidence:** 5

**Metareview:**

I fully agree with the reviewer that this is great and thorough work that could be presented at MIDL 2021. The methodological contribution is not that substantial - combining an off-the-shelf 2D DeepLapV3 with a "artefact type" class predictor. The code is already public and available as grand-challenge algorithm that's great! Unfortunately the data cannot be shared. I would encourage the authors to incorporate the additional baseline experiments that were suggested by the reviewer and look forward to their presentation at MIDL 2021.

---

### Decision · Program_Chairs · 2021-05-11

Accept (Poster)